# Acid–Base Balance, Blood Gases Saturation, and Technical Tactical Skills in Kickboxing Bouts According to K1 Rules

**DOI:** 10.3390/biology11010065

**Published:** 2022-01-02

**Authors:** Łukasz Rydzik, Mateusz Mardyła, Zbigniew Obmiński, Magdalena Więcek, Marcin Maciejczyk, Wojciech Czarny, Jarosław Jaszczur-Nowicki, Tadeusz Ambroży

**Affiliations:** 1Institute of Sports Sciences, University of Physical Education, 31-571 Cracow, Poland; tadek@ambrozy.pl; 2Department of Physiology and Biochemistry, Faculty of Physical Education and Sport, University of Physical Education, 31-571 Cracow, Poland; mmardyla93@gmail.com (M.M.); magdalena.wiecek@awf.krakow.pl (M.W.); marcin.maciejczyk@awf.krakow.pl (M.M.); 3Department of Endocrinology, Institute of Sport—National Research Institute, 01-982 Warsaw, Poland; zbigniew.obminski@insp.waw.pl; 4College of Medical Sciences, Institute of Physical Culture Studies, University of Rzeszow, 35-959 Rzeszow, Poland; wojciechczarny@wp.pl; 5Department of Tourism, Recreation and Ecology, University of Warmia and Mazury in Olsztyn, 10-719 Olsztyn, Poland

**Keywords:** acid–base balance, kickboxing, metabolic acidosis

## Abstract

**Simple Summary:**

The aim of our study was to analyze the changes in ABB after a three-round kickboxing fight and the level of technical and tactical skills presented during the fight. Fighting in kickboxing under K1 rules takes place with a high presence of anaerobic metabolism. Kickboxing athletes must have a good tolerance for metabolic acidosis and the ability to conduct an effective duel despite ABB disorders. Properly developed post-workout regeneration also plays an extremely important role.

**Abstract:**

Background: Acid–base balance (ABB) is a major component of homeostasis, which is determined by the efficient functioning of many organs, including the lungs, kidneys, and liver, and the proper water and electrolyte exchange between these components. The efforts made during competitions by combat sports athletes such as kickboxers require a very good anaerobic capacity, which, as research has shown, can be improved by administering sodium bicarbonate. Combat sports are also characterized by an open task structure, which means that cognitive and executive functions must be maintained at an appropriate level during a fight. The aim of our study was to analyze the changes in ABB in capillary blood, measuring levels of H^+^, pCO_2_, pO_2_, HCO_3_^−^, BE and total molar CO_2_ concentration (TCO_2_), which were recorded 3 and 20 min after a three-round kickboxing bout, and the level of technical and tactical skills presented during the fight. Methods: The study involved 14 kickboxers with the highest skill level (champion level). Statistical comparison of mentioned variables recorded prior to and after a bout was done with the use of Friedman’s ANOVA. Results: 3 min after a bout, H^+^ and pO_2_ were higher by 41% and 11.9%, respectively, while pCO_2_, HCO_3_^−^, BE and TO_2_ were lower by 14.5%, 39.4%, 45.4% and 34.4%, respectively. Furthermore, 20 min after the bout all variables tended to normalization and they did not differ significantly compared to the baseline values. Scores in activeness of the attack significantly correlated (r = 0.64) with pre–post changes in TCO_2_. Conclusions: The disturbances in ABB and changes in blood oxygen and carbon dioxide saturation observed immediately after a bout indicate that anaerobic metabolism plays a large part in kickboxing fights. Anaerobic training should be included in strength and conditioning programs for kickboxers to prepare the athletes for the physiological requirements of sports combat.

## 1. Introduction

Acid–base balance (ABB) is a major component of homeostasis, which is determined by the efficient functioning of many organs, including the lungs, kidneys, and liver [1,2], and the proper water and electrolyte exchange between these components [3]. At rest, an equilibrium is maintained between the pulmonary gas pressures (pO_2_ and pCO_2_) in the human body, and the buffering system in the blood consists of hydrogen ion (H^+^) acceptors, which include bicarbonate ions (HCO_3_^−^), proteins, amino acids, hydrogen phosphate ions (HPO_4_^2−^), and hemoglobin contained in erythrocytes. The buffering system compensates for small fluctuations in acidity, but intense exercise disrupts ABB in the body. Studies using 31P nuclear magnetic resonance spectroscopy have demonstrated a significant intracellular increase in H^+^ concentration and a concomitant decrease in phosphocreatine (PCr) concentration during some seconds of repeated supramaximal exercise [4]. With buffering, the post-exercise blood H^+^ concentration is much lower than in the cytosol. The peripheral and central chemoreceptors responsible for the regulation of the pulmonary ventilation rate respond to post-exercise changes in pH and disturbances in gas parameters of ABB [5,6]. Studies have shown that the higher the intensity of anaerobic exercise, the higher the blood levels of lactate and hydrogen ions and the greater the decrease in HCO_3_^−^ [7,8,9,10,11,12,13].

It is believed that the temporary high acidification of cytosol in muscle fiber cells is not the only cause of the decrease in maximal power [14]. Nevertheless, numerous studies have confirmed that an increase in blood buffering capacity after oral administration of sodium bicarbonate or other alkalizing fluids reduces post-exercise pH disturbances and increases exercise capacity [15,16,17,18,19,20,21].

Exercise during competitions in combat sports such as kickboxing, boxing, taekwondo, and wrestling requires very good anaerobic capacity, which, as demonstrated by studies, can be improved by administering sodium bicarbonate [22,23,24]. Combat sports are also characterized by an open task structure, which means that cognitive functions must be maintained at an appropriate level during a fight [25]. The following cognitive abilities have been most often studied in combat sport athletes because the levels of these features are in a great part related to athletic skills. The most frequently tested cognitive functions in athletes are visuo-motor coordination [26,27,28], information processing and planning [29,30], and accuracy of decision-making [31]. It should be noted that in hitting sports, such as boxing and kickboxing, testing of cognitive functions matters for assessment of brain micro-injuries among athletes [32,33]. In laboratory tests, measurements of the speed and accuracy of reactions to visual stimuli are used to evaluate the level of some of the above-mentioned skills, including simple reaction tests, choice reaction tests, GO/NOGO tasks, Stroop tests, and trail making tests. Many published results have indicated a relationship between the level of performance during these tests and the physiological responses induced by various laboratory efforts. Immediately after intense exercise-induced metabolic acidosis, the results of the reaction time test and Stroop test are worse than at rest, but after a 15 min rest, they partially normalize [34]. Combined with concurrent psychometric testing, exercise tests do not fully replicate the task structure of a typical real fight. In most cases, the scale of difficulty in completing a mental and physical task during official combat sports competitions is much greater compared to laboratory psychophysical tests.

The experiments conducted in these studies demonstrated bidirectional changes in the level of performance during psychometric tests in response to laboratory test exercises of increasing intensity. These are fundamental to understanding the relationships between exercise intensity and physiological responses with level of performance of psychometric tasks. During the first phase of low-intensity exercise, the choice reaction time progressively decreases until a blood lactate concentration of 5.5 mmol/L is reached, and progressively increases once this concentration is exceeded [35]. A similar biphasic pattern of choice reaction time has been recorded during running with increasing speed [36]. Furthermore, higher levels of cognitive function are presented by individuals with a higher physical capacity [37]. In the case of psychometric tests that examine simultaneously the speed and accuracy of reactions using the GO/NOGO test, there is a need to choose between two contradictory decisions, one favoring speed and the other oriented towards accuracy. It has been demonstrated that the choice between these options may depend on the task structure of the sport.

Studies have shown that karate athletes prefer a higher speed of response to a stimulus but make more mistakes, while rowers do the opposite [31]. In this study, both groups demonstrated improved performance on psychometric tests in subsequent attempts. This phenomenon is known as the effect of learning a response to the same repeated stimuli [38]. In the case of executive functions used during fighting with an opponent the athlete does not know yet, there is a very large variety of stimuli and many choices of responses to them, which minimizes the learning effect observed during repeated laboratory testing. For this reason, in addition to physical fitness, the outcome of the competition is determined by the level of technical and tactical skills.

There have been few attempts to numerically assess the level of specific executive functions as a technical skill. Three basic parameters that characterize the level of performance during a real kickboxing bout have been developed and implemented [32,33,39,40]. The literature to date lacks a comparison of the assessment of such technical skills with measurements of physiological responses during competitions in combat sports, with athletes using fist punches and/or kicks. Ass mentioned earlier, boxing, kickboxing and taekwondo athletes are exposed to head injuries during competitions, which can reduce cognitive and executive abilities and impair technical skills [41,42]. In addition, three repeated maximal physical efforts may contribute to physiological changes [43,44], which are, in part, responsible for accumulation of fatigue. The most visible symptom of increasing fatigue during successive bouts may be an increasing number of pauses and total time of pauses [45]. Although the analysis of the used types of offensive actions and their number during kickboxing matches have been presented in the literature, the novel skill parameters as a mirror of the levels of cognitive–executive function, together with complex physiological responses to the contest, are lacking. Thus, the aim of our study was to analyze the changes in ABB after a three-round kickboxing bout and the level of technical and tactical skills presented during the bout.

## 2. Materials and Methods

The study involved 14 kickboxers presenting the highest sports skill level (champion level).

The sports skill level was evaluated based on sporting achievements and having a kickboxing master’s degree, and the coach’s opinion. The minimum training experience of the subjects was between 8 and 10 years. The participants were aged 19 to 35 years. Details of the study group are presented in Table 1.

### 2.1. Analysis of the Fight

The competitors had one bout each according to K1 rules in the morning after two days of a training break. The fights were held according to the rules of the World Association of Kickboxing Organizations (WAKO) and consisted of three 2-min rounds separated by 1-min rests.

The fights were simulated, but took place on neutral ground and were refereed by a qualified referee. The athletes were matched in a manner consistent with their weight category.

The determination of the technical and tactical performance parameters was made based on video recordings of the bouts. Subsequent analysis was conducted using specialized formulas [40,46,47].

The efficiency of the attack indicates a number of scored points influencing the final result of the bout compared to the number of bouts observed.

Efficiency of the attack (S_a_)
$$S_{a}=\frac{n}{N}$$

*n*—number of attacks scoring 1 point.* In K1 rules, each clean hit of the opponent scores 1 point*N*—sum of observed bouts (*N* = 1 in this study)

The effectiveness of the attack denotes the number of scoring techniques compared to all the offensive actions performed.

Effectiveness of the attack (E_a_)
$$E_{a}=\frac{number\;of\;effective\;attacks}{number\;of\;all\;attacks}\;\times \;100\;$$

An effective attack is a technical action awarded a pointNumber of all attacks is a number of all offensive actions

The activeness of the attack describes the engagement of the athlete, indicating the number of offensive actions performed during the observed fights.

Activeness in the attack (A_a_)
$$A_{a}=\frac{number\;of\;all\;registered\;offensive\;actions\;of\;a\;kickboxer}{number\;of\;fights\;fought\;by\;a\;kickboxer\;\left(1\;in\;this\;study\right)}$$

### 2.2. Acid–Base Balance Analysis

ABB parameters were analyzed using an EPOC gasometer (Siemens, Ottawa, ON, Canada) immediately after 95 µL of arterialized fingertip blood was drawn into glass capillaries containing calcium-balanced lithium heparin (65 IU/mL). The determinations were made 5 min before the bout (measurement I), and 3 min (measurement II) and 20 min after the bout (measurement III).

Hydrogen ion concentration (H^+^), partial pressure of oxygen (pO_2_), and partial pressure of carbon dioxide (pCO_2_) were measured, and base excess in the extracellular fluid (BEecf) concentration of bicarbonate ions HCO_3_^−^ and TCO_2_ (total molar carbon dioxide concentration) were calculated.

### 2.3. Bioethics Committee

Prior to participation in the tests, the competitors were informed about the research procedures, which were in accordance with the ethical principles of the Declaration of Helsinki WMADH (2000). Obtaining the competitors’ written consent was the condition for their participation in the project. The research was approved by the Bioethics Committee at the Regional Medical Chamber (No. 287/KBL/OIL/2020).

### 2.4. Statistical Analysis

Statistica 13.1 software (StatSoft, Cracow, Poland) was employed for statistical analysis. Friedman’s ANOVA test was used to compare the results of repeated measures. The post-hoc test was Dunn’s test. Correlation analysis between selected variables was performed using Pearson’s linear correlation test. The Shapiro–Wilk test was used to test data for normal distribution. Effect size was calculated according to the formula:

Kendall’s W = Chi ^2/^N(K − 1), N = sample size, K = number of measurements. The level of statistical significance was set at *p* < 0.05.

## 3. Results

The biochemical indices studied changed significantly during the bout and recovery. The greatest changes were observed after the second measurement (3 min after the bout) (Table 2). For pCO_2_, the difference was significant only between measurements I and II, for H^+^, pO_2_, HCO_3_^−^ between measurements I and II and between measurements II and III, and for BE between all measurements (Table 2).

In most cases, the differences between measurements I and II were highest. They were also relatively high between measurements II and III, while the smallest (non-significant) differences were observed between measurements I and III (Table 2).

The recovery rate 20 min after the end of the bout was highest for H^+^ and was 96.97 ± 45.94%, whereas for the other ABB parameters the rate reached 68.57 ± 18.44% for HCO_3_^−^ and 68.30 ± 13.62% for BE.

The activeness of the attack was evaluated at a mean level of 96.9 ± 43.6 and the range of scores was from 68 to 198. For the efficiency of the attack, it was a mean score of 50.1 ± 12.8, and ranged from 37 to 79, whereas for the effectiveness of the attack, it was a mean of 54.5 ± 7.9, ranging from 39.9 to 64.5 (Table 3).

There were no significant correlations between the changes in the parameters induced by the bouts for [H^+^], pCO_2_, pO_2_, HCO_3_^−^, BE (ecf), or BE (b). The only positive correlation was found between molar concentrations of CO_2_ (TCO_2_) and the activity of the attack, which suggests that the greater the physical activity, the greater rise of CO_2_ concentration in the blood (Table 4).

## 4. Discussion

The significant changes in ABB parameters and blood oxygen and carbon dioxide saturation immediately after the bout indicate a large contribution of anaerobic metabolism in generating the physical work and gas exchange rate in kickboxers. The examinations conducted by other authors in kickboxers immediately after each of three rounds or after an entire bout showed a significant decline in muscle strength, and a progressive increase in blood lactate levels and heart rates [43,44,45,48]. In our study there were very weak correlations between activity in attacks and rise of hydrogen ion levels, but a significant positive link between activity and post bout increase of total CO_2_ level. That relationship may suggest relatively high mechanical efficiency in tested contestants, according to the model describing relationships between mechanical work output and metabolism [49]. As was mentioned earlier, kickboxing belongs to a family of hitting martial arts similar to Muay Thai, karate and taekwondo, where upper and lower limbs are engaged in offensive actions [50]. Practitioners of all these sports use the same techniques of punches and kicks. One of the most effective kicking techniques is the roundhouse kick, when it is performed correctly. Biomechanical factors of this kick have been explored and described in detail [51]. To date, the literature lacks data on ABB and gasometric studies after kickboxing bouts. Few and fragmented data on ABB and gasometry have been published after boxing fights [13]. This sport differs from kickboxing in its task structure and variety of attacks, but the requirements for general physical fitness, technical skills, and the type of attacks using the upper limbs are very similar. Bout intensity assessed based on lactate and post-exercise changes in magnitude and direction for ABB and gas saturation in boxers [13,22] are also consistent with our findings. We also conducted our ABB and gasometry observations during the short-term recovery period. The results showed that there were no statistically significant differences between baseline and post-exercise recovery measured at 20 min after the bout. However, the results showed that the normalization of parameters was not fully achieved, which indicates a deep disruption of homeostasis caused by the bout. Glycolytic metabolism exercise significantly decreases the levels of bicarbonate as a main factor in neutralizing hydrogen ions in the blood. This process occurs according to the following equation: H^+^ + HCO_3_^−^ → H_2_O + CO_2_ ↑. Despite the increased release of carbon dioxide into the blood, its saturation decreased during the bout due to increased gas exchange in the alveoli and intensifying hyperventilation [52,53]. Decreases in blood pCO_2_ and bicarbonate levels during intense exercise have also been reported by previous researchers. Similar post-exercise changes in ABB and blood gas parameters were noted in non-athletes and athletes, but a slight increase in oxygen saturation (by 14%) was observed only in athletes [54]. We found a similar (although slightly smaller, ca. 10%) increase in oxygen saturation after a kickboxing bout. This may suggest that regular physical training induces such an adaptive mechanism. It is important to mention that under resting conditions, hyperventilation is responsible for blood alkalization, since in the case of negligible lactate levels and the associated source of hydrogen ions, the main H ^+^ donor is the reversible reaction of CO_2_ + H_2_O ↔ HCO_3_^−^ + H^+^. It has been shown that short-term resting hyperventilation at a mean lactate concentration of 1.9 mmol/L leads to a decrease in the partial pressure of CO_2_ to a mean value of 21 mmHg and significant blood alkalization (pH = 7.6, H^+^ = 23 nmol/L). During competitions in combat sports, it is not possible to quantify the level of hyperventilation. We also did not measure the ventilation immediately before the bout. Hyperventilation attenuates the post-exercise decrease in pH, reduces CO_2_ saturation, and increases anaerobic power, especially at the end of a set of efforts [55]. A beneficial effect of pre-exercise hyperventilation before a competition on short-distance swimming performance has also been reported [56]. The reduction in CO_2_ saturation and hydrogen ion concentration due to hyperventilation improves physical performance during repeated resistance efforts [55]. Therefore, as documented, the beneficial effect of exercise-induced hyperventilation results from physiological responses. Blood alkalization obtained pharmacologically prior to graded exercise has been found to delay the onset of hyperventilation during work, which, according to researchers, confirms that lactate acidosis increases respiratory rate [57].

The appropriate level of cognitive functions is of great importance in open skill sports. The results of psychometric measurements, reaction time, and decision-making in female and male kickboxers have shown differentiation depending on sex and rules of competition (light vs. full contact) [58], but, to date, there has been no comparison of the results of laboratory psychometric tests with the assessment of executive functions, i.e., the technical performance during a real bout. The few studies conducted in kickboxers have only been designed to assess activeness during the bout. A slightly higher number of higher- and lower-intensity actions was demonstrated in light-category athletes [45] and a higher ratio of activity time to rest time was found in another study [59]. Noticeable differences have been reported in activity and offensive fighting style in winners and losers based on the number of combined punches, kicks, and alternating hand and leg actions [60].

## 5. Conclusions

The disturbances in ABB and changes in blood oxygen and carbon dioxide saturation observed immediately after a bout indicate that anaerobic metabolism plays a large part in kickboxing fights. Anaerobic training should be included in strength and conditioning programs for kickboxers to prepare the athletes for the physiological requirements of sports combat.K1 kickboxers must be characterized by good metabolic acidosis tolerance and the ability to fight effectively despite ABB disturbances, and show good post-exercise recovery.

## Figures and Tables

**Table 1 biology-11-00065-t001:** Anthropometric measures of study participants.

Variables	No	M	Me	Min	Max	Q1	Q3	SD
Body mass	14	84.90	85.50	75.00	90.00	83.00	88.50	4.93
Body height	14	181.05	180.00	175.00	189.00	179.00	183.50	3.39
BMI	14	26.04	25.99	24.12	28.64	25.15	26.73	1.24

No—number, M—mean, Me—median, Min—minimum, Max—maximum, Q1—first quartile, Q3—third quartile, SD—standard deviation.

**Table 2 biology-11-00065-t002:** The level of acid–base balance parameters in the tested group of athletes in three consecutive measurements.

Parameter	Measurement									Friedman’s ANOVA		Post-Hoc (Dunn’s Test)	Effect Size
	I (*n* = 14)			II (*n* = 14)			III (*n* = 14)						
	M	Me	SD	M	Me	SD	M	Me	SD	Chi^2^	*p*	I-II	I-II
H^+^ (nmol/L)	37.9	37.0	3.3	54.0	49.0	9.8	41.1	40.0	3.9	22.29	<0.001	<0.05	0.80
pCO_2_ (mmHg)	37.2	37.3	3.3	31.8	31.9	2.6	35.2	35.0	0.7	7.43	0.024	<0.05	0.27
pO_2_ (mmHg)	77.2	75.2	6.0	85.6	85.1	8.5	73.9	75.8	4.5	16.15	<0.001	<0.05	0.58
HCO_3_^−^ (mmol/L)	24.6	25.3	1.3	14.9	15.4	1.6	21.3	21.6	1.8	24.57	<0.001	<0.05	0.88
BE mmol/L	0.5	0.9	1.2	−11.9	−10.6	2.7	−3.7	−3.2	2.4	28.00	<0.001	<0.05	1.00
TCO_2_(mmol/L)	24.1	25.1	1.3	15.8	16.1	1.4	21.5	21.7	1.1	24.50	<0.001	<0.05	0.88

M—mean, Me—median, SD—standard deviation. NS—not statistically significant, I—before exercise, II—3 min after exercise, III—20 min after exercise.

**Table 3 biology-11-00065-t003:** Value of activeness, efficiency, and effectiveness of the attack.

Variables	No	M	Me	Min	Max	Q1	Q3	SD
Activeness of the attack	14	96.9	79.0	68.0	198.0	76.0	96.0	43.6
Efficiency of the attack	14	50.1	47.0	37.0	79.0	45.0	49.0	12.8
Effectiveness of the attack	14	54.5	54.4	39.9	64.5	49.0	60.8	7.9

**Table 4 biology-11-00065-t004:** Matrix of correlation coefficients between examined variables.

Variables		Activeness of the Attack		Efficiency of the Attack		Effectiveness of the Attack	
		R	*p*	R	*p*	R	*p*
∆ = I-II	[H^+^]	0.11	0.62	0.07	0.81	0.07	0.808
	pCO_2_ (mmHg)	0.14	0.14	−0.03	0.90	0.00	1.00
	pO_2_ (mmHg)	0.14	0.62	−0.32	0.26	−0.03	0.90
	HCO_3_^−^ (mmol/L)	−0.21	0.46	−0.25	0.38	0.32	0.26
	BE (ecf) mmol/L	−0.10	0.71	−0.01	0.95	−0.41	0.13
	TCO_2_ mmol/L	0.64	0.01	−0.32	0.26	−0.17	0.54

Valuesin bold are statistically significant

## Data Availability

The data presented in this study are available on request from the corresponding author.

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
