# Peer review of "Acid–Base Balance, Blood Gases Saturation, and Technical Tactical Skills in Kickboxing Bouts According to K1 Rules"

_biology, 2022, doi:10.3390/biology11010065_

Round 1

Reviewer 1 Report

Acid-base balance, blood gases saturation, and technical tactical 2

skills in kickboxing bouts according to K1 rules

The authors did not change nothing that I asked. I'm afraid that the version sent was not updated. Also, I strongly suggest to authors reply my comments indicating clearly where are the changes (line, page).

Author Response

Dear Reviewer,

Thank you very much for your time and valuable comments, which all have been considered and incorporated. The detailed list of responses is given below. We hope that the modifications and explanation will be acceptable for you.

Yours sincerely,

Rydzik, corresponding author

Main comments The theme of acid-base balance during exercise is very important because refers to a most important factor of whole-body homeostasis. During exhaustive exercise as combat sports, large amounts of lactate accumulate in both muscle and blood, and the corresponding accumulation of proton ions creates metabolic acidosis. The authors aimed to analyze the alterations in acid-base balance after a 3-round kickboxing bout/fight and try to relate it to level of tactical and technical skills demonstrated during the fight. 14 kickboxers were enrolled in the study. The biochemical responses were changed significantly during the bout and during recovery. The article is interesting and has the following strengths: 1) the complete analysis of the acid-base balance system, 2) the practical nature of the evaluation during the fights, 3) the good discussion with the sports literature.

            The weaknesses are only in minor aspects, such as

1) adding specific results (with statistics) to the abstract and

A: This has been corrected in lines 27-33

 Better defining the concepts of efficiency and effectiveness.

A: This has been corrected :

Effectiveness - lines 149-150

Attack Effectiveness - lines 158-159

Attack Activity - lines 166-167

 3) informing perspectives from the study, relating for example, biomechanical studies trying to associate the metabolic impact against changes in biomechanical parameters of specific movements in combats as kicks (for example, https://pubmed.ncbi.nlm.nih.gov/28841670/ and https://pubmed.ncbi.nlm.nih.gov/29952865/ )

A: Added more information lines 253-264

The text is well written and the tables are well structured.

A: Thank you

Minor points Line 46 - several is too much, consider using some .

A: Line 51 has been corrected

Lines 62-64 - consider including references to support these statements.

A: Lines 67-73 have been corrected

Line 117 - body mass instead of body weight.

A: Corrected In Table 1

 Line 132 - please, consider using the term absolute effectiveness instead of efficiency (I suggest read this review paper on these concepts: https://doi.org/10.3389/fphys.2018.01789 , perhaps to justify the nomenclature used). And the following effectiveness (divided by total attacks) can be relative or percentage effectiveness.

A: Dear reviewer, this is a generic term introduced by us and published and reviewed in many papers. It is the result of analysis of earlier works that used similar terminology

  1. Rydzik, Ł.; Niewczas, M.; Kędra, A.; Grymanowski, J.; Czarny, W.; Ambroży, T. Relation of indicators of technical and tactical training to demerits of kickboxers fighting in K1 formula. Arch. Budo Sci. Martial Arts Extrem. Sport. 2020, 16, 1–5.
  2. Rydzik, Ł.; Maciejczyk, M.; Czarny, W.; Kędra, A.; Ambroży, T. Physiological Responses and Bout Analysis in Elite Kickboxers During International K1 Competitions. Front. Physiol. 2021, 12, 737–741, doi:10.3389/fphys.2021.691028.
  3. Rydzik, Ł.; Ambroży, T. Physical fitness and the level of technical and tactical training of kickboxers. Int. J. Environ. Res. Public Health 2021, 18, 1–9, doi:10.3390/ijerph18063088.
  4. Błach, W.; Ouergui, I.; Rydzik, Ł.; Ambro, T. Evaluation of the Body Composition and Selected Physiological Variables of the Skin Surface Depending on Technical and Tactical Skills of Kickboxing Athletes in K1 Style. 2021.
  5. Ambroży, T.; Rydzik, Ł.; Obmiński, Z.; Klimek, A.T.; Serafin, N.; Litwiniuk, A.; Czaja, R.; Czarny, W. The Impact of Reduced Training Activity of Elite Kickboxers on Physical Fitness, Body Build, and Performance during Competitions. Int. J. Environ. Res. Public Health 2021, 18, 4342, doi:10.3390/ijerph18084342.

Take care on line 37,. Line 159 - space before parenthesis.

A: Line 186 has been corrected

Lines 206-207 - consider citing the source of ‘few and fragmentary data”.

A:Completed line 266,

Reviewer 2 Report

Contributions made to researchers have been responded to and modified.

Author Response

Dear Reviewer,

Thank you for your review, which certainly raised the quality of our manuscript.

Yours sincerely,

Rydzik, corresponding author

Reviewer 3 Report

Line 233-234. "The only positive correlation was found between
molar concentrations of CO2 (TCO2) and the activity of the attack, which suggests that the greater the physical activity {or effort?}, the greater the decrease in this parameter". Was this point thoroughly discussed in the Discussion section; i don't seem to be able to find it?

Author Response

Dear Reviewer,

Thank you very much for your time and valuable comments, which all have been considered and incorporated. The detailed list of responses is given below. We hope that the modifications and explanation will be acceptable for you.

Yours sincerely,

Rydzik, corresponding author

Line 233-234. "The only positive correlation was found between
molar concentrations of CO2 (TCO2) and the activity of the attack, which suggests that the greater the physical activity {or effort?}, the greater the decrease in this parameter". Was this point thoroughly discussed in the Discussion section; i don't seem to be able to find it?

A: Thank you we have added this to our discussion in lines 253-264

Round 2

Reviewer 1 Report

          The weaknesses are only in minor aspects, such as

1) adding specific results (with statistics) to the abstract and

A: This has been corrected in lines 27-33

2nd Round: Ok, just consider using dot instead of comma as decimal separator. Also, to all correlation coefficient is needed the corresponding significance index (line 32).

 Better defining the concepts of efficiency and effectiveness.

A: This has been corrected :

Effectiveness - lines 149-150

Attack Effectiveness - lines 158-159

Attack Activity - lines 166-167

2nd Round: OK.

3) informing perspectives from the study, relating for example, biomechanical studies trying to associate the metabolic impact against changes in biomechanical parameters of specific movements in combats as kicks (for example, https://pubmed.ncbi.nlm.nih.gov/28841670/ and https://pubmed.ncbi.nlm.nih.gov/29952865/ )

A: Added more information lines 253-264

2nd Round: OK.

The text is well written and the tables are well structured.

A: Thank you

Minor points Line 46 - several is too much, consider using some .

2nd Round: OK.

A: Line 51 has been corrected

2nd Round: OK.

Lines 62-64 - consider including references to support these statements.

A: Lines 67-73 have been corrected

2nd Round: OK.

Line 117 - body mass instead of body weight.

A: Corrected In Table 1

2nd Round: OK.

 Line 132 - please, consider using the term absolute effectiveness instead of efficiency (I suggest read this review paper on these concepts: https://doi.org/10.3389/fphys.2018.01789 , perhaps to justify the nomenclature used). And the following effectiveness (divided by total attacks) can be relative or percentage effectiveness.

A: Dear reviewer, this is a generic term introduced by us and published and reviewed in many papers. It is the result of analysis of earlier works that used similar terminology

  1. Rydzik, Ł.; Niewczas, M.; Kędra, A.; Grymanowski, J.; Czarny, W.; Ambroży, T. Relation of indicators of technical and tactical training to demerits of kickboxers fighting in K1 formula. Arch. Budo Sci. Martial Arts Extrem. Sport. 2020, 16, 1–5.
  2. Rydzik, Ł.; Maciejczyk, M.; Czarny, W.; Kędra, A.; Ambroży, T. Physiological Responses and Bout Analysis in Elite Kickboxers During International K1 Competitions. Front. Physiol. 2021, 12, 737–741, doi:10.3389/fphys.2021.691028.
  3. Rydzik, Ł.; Ambroży, T. Physical fitness and the level of technical and tactical training of kickboxers. Int. J. Environ. Res. Public Health 2021, 18, 1–9, doi:10.3390/ijerph18063088.
  4. Błach, W.; Ouergui, I.; Rydzik, Ł.; Ambro, T. Evaluation of the Body Composition and Selected Physiological Variables of the Skin Surface Depending on Technical and Tactical Skills of Kickboxing Athletes in K1 Style. 2021.
  5. Ambroży, T.; Rydzik, Ł.; Obmiński, Z.; Klimek, A.T.; Serafin, N.; Litwiniuk, A.; Czaja, R.; Czarny, W. The Impact of Reduced Training Activity of Elite Kickboxers on Physical Fitness, Body Build, and Performance during Competitions. Int. J. Environ. Res. Public Health 2021, 18, 4342, doi:10.3390/ijerph18084342. 
  1. 2nd Round: OK.

Take care on line 37,. Line 159 - space before parenthesis.

A: Line 186 has been corrected

2nd Round: OK.

Lines 206-207 - consider citing the source of ‘few and fragmentary data”.

A:Completed line 266,

2nd Round: OK.